# Mask-Moving-Lithography-Based High-Precision Surface Fabrication Method for Microlens Arrays

**DOI:** 10.3390/mi15020289

**Published:** 2024-02-19

**Authors:** Jianwen Gong, Ji Zhou, Junbo Liu, Song Hu, Jian Wang, Haifeng Sun

**Affiliations:** 1Institute of Optics and Electronics, Chinese Academy of Sciences, Chengdu 610209, China; gongjw@ioe.ac.cn (J.G.); zhouji@ioe.ac.cn (J.Z.); ljbopt@126.com (J.L.); husong@ioe.ac.cn (S.H.); wangjian@ioe.ac.cn (J.W.); 2School of Optoelectronic Science and Engineering, University of Electronic Science and Technology of China, Chengdu 610054, China; 3University of Chinese Academy of Sciences, Beijing 100049, China

**Keywords:** microlens arrays, pre-exposure technology, inverted air bath reflux method

## Abstract

Microlens arrays, as typical micro-optical elements, effectively enhance the integration and performance of optical systems. The surface shape errors and surface roughness of microlens arrays are the main indicators of their optical characteristics and determine their optical performance. In this study, a mask-moving-projection-lithography-based high-precision surface fabrication method for microlens arrays is proposed, which effectively reduces the surface shape errors and surface roughness of microlens arrays. The pre-exposure technology is used to reduce the development threshold of the photoresist, thus eliminating the impact of the exposure threshold on the surface shape of the microlens. After development, the inverted air bath reflux method is used to bring the microlens array surface to a molten state, effectively eliminating surface protrusions. Experimental results show that the microlens arrays fabricated using this method had a root mean square error of less than 2.8%, and their surface roughness could reach the nanometer level, which effectively improves the fabrication precision for microlens arrays.

## 1. Introduction

Microlens arrays can achieve high integration and superior optical performance of optical systems due to their small size, light weight, and high integration. Therefore, they have been widely used in applications such as artificial intelligence, fiber optics, laser radar, information processing, aerospace, biomedical, and laser technology [1,2]. As typical micro-optical elements, the optical performance of microlens arrays determine the performance of highly integrated optical or optoelectronic systems. In addition, as two-dimensional (2D) refractive optical elements, microlens arrays deliver high optical performance in wavefront measurement, three-dimensional (3D) imaging, illumination light field homogenization, and beam shaping [3]. With advances in information technology, the increasing integration of microlens arrays imposes more stringent requirements on the fabrication precision for unit lenses. The continuous reduction in the sizes of unit lenses makes it difficult to control surface shape errors and surface roughness, leading to a decline in the optical performance of microlens arrays. Therefore, reducing the surface shape errors and surface roughness of microlens arrays is a prerequisite to ensure their superior optical performance.

As micro/nano structural components with continuous surfaces, fabrication technologies for microlens arrays include laser direct writing [4,5,6], multi-layer etching [7,8], grayscale lithography [9,10,11], 3D printing technology [12,13,14], and mask-moving technologies [15]. Direct writing technology offers a high precision and resolution but is costly and less efficient, so it is unsuitable for mass fabrication or fabrication of large-sized micro-optical elements. Multi-layer etching technology can only approximate continuous-surface microstructures with multiple steps and is mainly used to fabricate focusing lens with lower requirements. Moreover, as the number of steps increases, the need for repeating the transfer of the etched pattern results in high costs, and achieving alignment accuracy and surface precision becomes challenging. In addition, grayscale lithography technology involves complex designs of grayscale masks for different microlens shapes, leading to high fabrication costs.

Mask-moving technology [16], as an efficient 3D micro/nanostructure fabrication method, allows for continuous modulation of the exposure dose by moving the binary mask’s position during the exposure process, thereby enabling the fabrication of 2D and 3D micro/nanostructures. Cao Axiu et al. conducted research on miniaturized compound eye structures and designed a novel compound eye imaging structure for multi-dimensional information detection [17]. Shi Lifang et al. proposed an effective exposure method for aspherical microlens arrays based on an aperiodic moving mask [18,19,20,21]. Dong Xiaochun et al. [22] proposed a mask-moving-based fabrication method using contact and proximity lithography equipment, dividing the 3D target structure into numerous strip areas and projecting the microstructures within each strip area to generate 2D mask sub-functions. This method extends the range of fabrication from simple, regular micro-relief structures to arbitrary continuous-surface microstructures, addressing the fabrication challenges of micro-optical elements with special surface shapes and unique arrangements.

The mask-moving-based lithography technology, as shown in Figure 1, mainly employs a contact and proximity exposure system, maskless digital lithography and mask projection lithography. Among them, the near-contact lithography mainly adopts the near-contact exposure system. As shown in Figure 1a, its limitations in principle lead to a limited minimum diameter of the microlens and a low precision of the machining surface profile, which is not suitable for mass production and processing. The optical projection lithography technology [23,24] transfers the pattern on the mask to the photoresist on the silicon wafer surface in a step-by-step or scanning-exposure manner based on the optical projection imaging principle. This technology is categorized into digital maskless lithography and mask-based projection lithography, as shown in Figure 1b,c. With an objective lens system with magnification, it scales down the feature pattern to the substrate surface, fulfilling the high-precision requirements for micro/nano devices. Digital maskless lithography technology usually involves layer-by-layer slicing and multiple exposure processes, with the feature linewidth of the exposure pattern depending on the size of the micro-mirror unit. It cannot ensure high precision and large-area exposure requirements [25,26]. As the mainstream technology for fabrication of advanced-node micro/nanostructures, mask-based projection lithography technology offers advantages such as a large exposure area, high precision, and efficiency, making it suitable for high-precision fabrication of microlens arrays [27,28]. On this basis, we propose a moving-mask lithography method based on projection, which adopts a 0.2 times projection objective to improve the resolution of line width and reduce the difficulty of mask processing [29]. However, due to the continuous reduction in the size of the unit lens, it is difficult to control the surface shape error and surface roughness, resulting in the degradation of the optical performance of the microlens array composed of the unit lens. Therefore, how to reduce the surface error and surface roughness of the microlens array is the premise of ensuring its superior optical performance.

In this study, a mask-moving-projection-lithography-based high-precision surface fabrication method for microlens arrays is proposed. This method combines the moving mask and projection lithography technologies for high-precision fabrication of microlens arrays. The root mean square error of the surface shape is less than 2.8%, and the surface roughness can reach the nanometer level. To ensure the precision of the microlens array’s surface profile, pre-exposure technology is used to increase the photoresist development speed and reduce the photoresist threshold. In addition, the inverted air bath reflux method is used to heat the exposed photoresist in the microlens array’s surface area to a molten state. Under the influence of surface tension, it eliminates irregular protrusions on the microlens array’s surface, thus reducing surface roughness. Finally, experiments were conducted on typical hexagonal compound eye lenses and square microlenses. The results indicate that this method offers higher efficiency in microlens array fabrication and can achieve high-precision surface fabrication for microlens arrays.

## 2. Methods

### 2.1. Principle of Projection Lithography Based on Mask Movement

A projection lithography exposure system projects the mask feature pattern onto the substrate surface with a reduced magnification. Figure 2 illustrates the principle of mask movement based on a projection lithography system. A substrate coated with the photoresist is fixed on the 2D moving worktable of the projection exposure system, while the mask is placed on the mask plane of the projection exposure system. The working principle of the mask movement projection exposure system is shown in Figure 2a. The exposed image on the substrate surface is the same as the mask feature pattern, but with reduced dimensions, as shown in Figure 2. As the substrate moves translationally during the exposure process, the substrate surface is scanned through exposure openings, achieving a continuous exposure distribution of the feature pattern. The mask version is shown in Figure 2b. When the target structure is equipartition interval is sufficient, the relief structure in the small area is approximately cylindrical. During the exposure process, the mask moves at a constant speed along the objective function equipartition direction, and the exposure distribution generated by the mask coding unit is superimposed to form the exposure distribution on the surface of the resist, as shown in Figure 2c.

This method uses one-dimensional mask movement for a single exposure, enabling the fabrication of various complex 3D nanostructures with intricate surface shapes. Additionally, the projection objective lens system with reduced magnification makes it easier to prepare mask templates under the same microlens aperture conditions and to fabricate microlens arrays with a low linewidth. The mask feature pattern is divided into a series of grayscale patterns, making the profile of the microlens array equivalent to a series of profile functions, which reduces the fabrication complexity. During this process, the 3D nanostructure is equally divided into multiple narrow strip areas, as shown in Figure 3a, and each strip area is encoded. When each sub-area of the target structure is narrow enough, these narrow areas can be approximated as cylindrical, and the 3D nanostructure can be represented by a mask-moving filtering function.

### 2.2. Pre-Exposure Principle

The mask-moving technology makes it possible to apply a complex exposure distribution to the resist surface by encoding the target structure. Precise control of the exposure distribution and dose of the photoresist is necessary to achieve high-precision control of the surface shapes of specific 2D and 3D micro/nanostructures. As the carrier medium for the feature pattern, the sensitivity of the photoresist to the exposure beam is an important indicator of the lithography process. Typically, the absorption of the photosensitive compounds to the light beam in the photoresist is similar to Lambert’s law [30]:(1)$$\frac{dI}{dz}=-\alpha \times z,$$
where *I* refers to the intensity of the light field in the longitudinal area within the photoresist layer at a depth of *z*, and α refers to the photon absorption coefficient of the photoresist. In the photoresist layer, the photon absorption coefficient α refers to a constant independent of the thickness of the photoresist layer and can be expressed by the following equation in the electromagnetic field theory:(2)$$\alpha =\frac{4\pi k}{\lambda },$$
where *k* refers to the imaginary part of the refractive index of the photoresist layer, and *λ* refers to the exposure wavelength. Therefore, integrating Equation (2) yields the following equation:(3)$$I\left(z\right)=I_{0}\times e^{-\frac{4\pi kz}{\lambda }},$$
where *I*_0_ refers to the initial intensity of the light field in the surface layer of the photoresist at *z* = 0. Equation (3) means that the intensity of the light field inside the photoresist layer follows an exponential relationship with the depth *z* when the intensity of the light field is kept constant. Usually, due to the influence of the photoresist’s properties and relevant process parameters, the relationship between exposure depth and exposure dose is not linear. Although the nonlinearity in the relationship between exposure depth and exposure dose of different photoresists varies, they generally follow the trend of a logarithmic function, which can be summarized as follows:(4)$$\gamma =\frac{1}{\ln E_{cl}-\ln E_{th}}=\frac{h\left(x,y\right)/H_{\max}}{\ln E\left(x,y\right)-\ln E_{th}},$$
where: $0<h\left(x,y\right)<H_{\max}$ $E_{th}<E\left(x,y\right)<E_{cl}$. γ refers to relative curvature and is expressed by a constant. $E\left(x,y\right)$ refers to the exposure dose required to achieve the exposure depth $h\left(x,y\right)$. $E_{cl}$ refers to the exposure dose required for all photoresist layers to react. $E_{th}$ refers to the minimum exposure dose (exposure threshold) at which the photochemical reaction begins in the photoresist surface. Derivation of Equation (4) yields the following equation:(5)$$E\left(x,y\right)=\exp\left(\frac{\gamma \times H_{\max}}{h\left(x,y\right)}\right)+E_{th}.$$

Equation (5) illustrates the mathematical relationship between the depth of the photoresist layer where the photochemical reaction occurs and the exposure dose, from which we can deduce the exposure depths of the photoresist layer at different exposure doses and thereby create an exposure dose model for the designed structure, as shown in Figure 4.

To verify the photosensitive and developing properties of the photoresist, a mobile exposure method was used to apply a linearly increasing exposure dose to the surface of the AZ4620 photoresist [31]. After development and cleaning, the exposure depths at different exposure doses were measured using a step profiler, and the measurement results are shown in Table 1. The curve fitting of the two parameters is shown in Figure 5.

As shown in Figure 5, as the exposure dose increases from the initial E_0_ to E, the exposure depth gradually increases. When the exposure dose reaches E, the photoresist’s exposure depth is 11.06 μm. When the exposure dose is less than E_0_, the photoresist’s exposure depth is 0, indicating the existence of an exposure threshold E_0_ for the photoresist. Therefore, when the exposure dose exceeds the exposure threshold, the photoresist undergoes a photo-degradation reaction. In the microlens fabrication process, the photoresist does not undergo any decomposition in areas with an exposure dose less than the exposure threshold, leading to severe deformation in those areas. To fabricate a microlens with a continuous surface, it is usually necessary to apply exposure doses ranging from 0 to E to the photoresist surface. The value of E is determined by the structural parameters of the microlens. To eliminate the impact of the exposure threshold on the surface shape of the microlens, the photoresist development speed is increased using the substrate pre-exposure method, which enhances the development substrate and achieves linear adjustment of the exposure surface shape.

### 2.3. Inverted Air Bath Reflux Method

Although the substrate pre-exposure method can control the shape and profile of the microlens, this method divides the 3D micro/nanostructure into narrow strip areas, encodes the microstructures in each strip area, and approximates these small areas as cylindrical, which results in step-like microstructures on the surface of the fabricated microlens, leading a high degree of surface roughness. Therefore, the inverted air bath reflux method is used in this study to control the surface roughness of the microlens. This method is a surface-shaping technique that heats the hemispherical micro/nanostructure pattern generated by lithography to a temperature above the glass transition temperature, allowing the photoresist, so as to form a lens shape with the pattern aperture as the boundary under the influence of surface tension. Additionally, this method avoids the uneven softening of the glass inside the lens caused by conventional heat conduction, ensuring the diameter and sag height of the microlens base.

With the conventional heating reflux method, as shown in Figure 6, when the temperature of the photoresist is higher than the glass softening point and the system reaches equilibrium, the gravity can be neglected in the mechanical analysis in Figure 6a based on the Bond number theory [32]. There exists an equilibrium equation *P*_i_ = *P*_a_ + *P*_s_, where *P*_a_, *P*_i_, and *P*_s_ refer to the internal pressure of the photoresist, atmospheric pressure, and the additional pressure in the vertical direction due to surface tension, respectively. Since *P*_a_ and *P*_s_ have the same direction of force, *P*_i_ is greater than *P*_s_. This indicates that with the passage of time, the profile height will gradually increase when the conventional heating reflux method is used, which will affect the surface shape precision of the microlens. As shown in Figure 6b, inverted aspherical microlens structures are placed in an oven where their surface precision is improved by controlling the temperature and time of the heating reflux. The developed microlens structures are placed in the oven for post-baking and melting. Post-baking makes the photoresist adhere more tightly to the substrate. Upon completion of post-baking, the oven temperature is increased for melting to reduce the surface roughness of the photoresist microlens. The surface of the photoresist microlens is heated to a molten state using the air bath reflux method, and the irregular protrusions on the microlens surface are eliminated under the influence of surface tension.

With the inverted air bath reflux method, since the gravity direction in the mechanical model points from the substrate to the surface of the microlens, which is opposite to the gravity direction in the conventional heating reflux method, the influence of gravity cannot be neglected. The analysis of the mechanical model in Figure 6b yields the following equilibrium equation:(6)$$P_{i}=P_{a}+P_{s}-G$$

In the initial stage, as a part of the microlens surface softens, the influence of gravity is greater than the influence of surface tension, i.e., $P_{s}$ < G. Therefore, it can be concluded that $P_{i}$ < $P_{a}$, indicating that with the increase in temperature in the initial stage, the thickness of the photoresist microlens will decrease. As the temperature continues to rise, the influence of gravity becomes less than the influence of surface tension, i.e., $P_{s}$ > G. Therefore, $P_{s}$ > $P_{a}$ indicates that the height of the photoresist microlens will gradually increase during the reflux process until it reaches equilibrium. This method can effectively control the height variation during the reflux process, ensuring the surface shape precision of the microlens.

## 3. Experiment and Analysis

The method proposed in the study uses a projection lithography system for the fabrication of microlenses. This system mainly consists of various subsystems such as an illumination system, projection objective lens, alignment system, focus detection system, worktable system, electronic control system, and software system (I700v-1). Its technical specifications include an effective field of view of 15 mm × 15 mm, a numerical aperture of 0.35, and an overall optical magnification of M = 1/5×, with a linewidth resolution better than 0.8 μm. To validate the feasibility of the experimental results of this method, two typical microlens array patterns were fabricated: a hexagonal compound eye lens with a diameter of 480 µm and a sag height of 2.5 µm, and a square microlens with a diameter of 603 µm and a sag height of 2 µm. Their surface profile precision and surface roughness were measured using a DektakXT step meter and 3D topography meter developed by Chinese Academy of Sciences photoelectric technology.

As shown in Figure 7a–d and Figure 8a–d, two different methods, i.e., conventional exposure and frontal substrate pre-exposure methods, were used to fabricate two types of microlenses. The surface shapes of the microlenses were measured using a step profiler. Among them, the photoresist was AZ9260, the adhesive thickness was 3.8 um, the conventional exposure dose was 78 mj/cm², the moving speed was 0.825 um/s, and according to the experimental results in Figure 5, the pre-exposure dose of the front base pre-exposure method was 12 mw/cm². As shown in Figure 7a–d and Figure 8a–d, the depths of the microlenses fabricated using both methods are approximately the same, but their surface shapes differ significantly.

The front base pre-exposure method avoids the influence of exposure threshold on the surface shape of the microlens, so the obtained surface shape loss is small, the surface shape profile of the lens surface is continuous, and there is no abrupt change in the profile shape, as shown in Figure 7b,d and Figure 8b,d. For the area where the exposure is less than the exposure threshold, the resist does not decompose in the microlens fabrication process, resulting in serious deformation in this area. The high-frequency loss of surface shape obtained by conventional lithography is very large, and a “flat top” appears at the top of the lens, as shown in Figure 7a and Figure 8a. The outline of the top of each lens is a horizontal line. The middle black area on the lens microscopic image also shows the phenomenon of “flat top” at the top of the lens, as shown in Figure 7c and Figure 8c, which seriously affects the surface shape of the lens. Measurements show that the root mean square error of the surface shape fabricated using the frontal substrate pre-exposure method is less than 2.8%, while that fabricated using the conventional lithography method is less than 11%. Therefore, the frontal substrate pre-exposure method with a moving mask can fabricate microlenses with higher evenness, lower high-frequency losses, and more complete surface shapes.

As shown in Figure 7b,d and Figure 8b,d, although the substrate pre-exposure method can control the shape and profile of the microlens, this method divides the 3D micro/nanostructure into narrow strip areas, encodes the microstructures in each strip area, and approximates these small areas as cylindrical, which results in step-like microstructures on the surface of the fabricated microlens, leading a high degree of surface roughness.

Figure 9a,b and Figure 10a,b, are the measurement results of the step height profiler of the two microlenses, respectively. In order to avoid the influence of the photoresist exposure threshold and thus the surface shape distortion in Figure 7a and Figure 8a, the method of pre-exposure with front base is adopted in the preparation of the two lenses. It can be seen that the uniformity of the two lenses is better, the high-frequency loss is less, there is no “flat top”, and the surface shape is complete. Since the mask plate pattern is a series of fine strip areas of equidistant slices, and each tiny area is approximately cylindrical, it can be seen from the local magnification images in Figure 9a and Figure 10a that the surface of the lens without surface heating and reflow has a stepped micro-structure, and the contour lines show a sawtooth shape, and the surface is rough and uneven with many burrs. As shown in Figure 9b and Figure 10b, the photoresist microlens surface is heated to a molten state by the method of gas bath heating. The hot reflux temperature was set at 140 ℃ and the hot reflux time was set at 80s. Under the action of surface tension, the photoresist on the lens surface becomes smooth and neat, the roughness is greatly reduced, and the cross-section profile is closer to the design profile.

Figure 11 and Figure 12 are the corresponding 3D topography profilometer scanning results of the two microlenses in Figure 9 and Figure 10, respectively. It can be intuitively seen from Figure 11a and Figure 12a that the surface of the lens without surface heating reflux is rough and has more burrs, which is consistent with Figure 7d and Figure 8d. Figure 11b and Figure 12b show the morphology measurements with the inverted air bath heating reflux method being used after exposure and development. In order to further explain the degree of influence, a three-dimensional topography measuring instrument was also used to measure the roughness, and it was found that the roughness changed from 96.8 nm to 13.6 nm, as shown in Figure 11a,b, and 88.6 nm to 9.5 nm, as shown in Figure 12a,b. Therefore, this method can effectively improve the contour surface accuracy without losing the contour shape accuracy, so the method is expected to be used in the fabrication of continuous surface structures and high-quality micro-optical components.

## 4. Conclusions

In this study, a mask-moving-based high-precision surface fabrication method for microlens arrays is proposed, which effectively reduces the surface shape errors and surface roughness of microlens arrays. Using a projection lithography system to achieve a 5× reduction in the transfer of feature patterns to the substrate reduces the mask preparation costs and improves the fabrication efficiency. By using the development threshold of the photoresist and combined with the pre-exposure technology to increase the development speed, the impact of the exposure threshold on the surface shape of the microlens is eliminated, thus achieving high-precision control of the surface profile of the microlens. During the development process, the photoresist microlens surface is heated to the molten state by using the inverted air bath reflux method, which reduces irregular protrusions on its surface and improves its surface roughness. To validate the feasibility of the experimental results of this method, two typical microlens array patterns were fabricated: a hexagonal compound eye lens with a diameter of 480 µm and a sag height of 2.5 µm, and a square microlens with a diameter of 603 µm and a sag height of 2 µm, and their surface shape precision and surface roughness were measured.

Measurements show that the root mean square error of the surface shape fabricated using the frontal substrate pre-exposure method is less than 2.8%, while that fabricated using the conventional lithography method is less than 11%. Therefore, the frontal substrate pre-exposure method with a moving mask can fabricate microlenses with higher evenness, lower high-frequency losses, and more complete surface shapes. The microlens fabricated using the inverted air bath reflux method has a smooth and even surface and significantly reduced roughness, and its cross-sectional profile matches the design shape better. The degrees of surface roughness of the microlenses fabricated using the inverted air bath heating reflux method are 13.6 nm and 9.5 nm, respectively, which are approximately 10 times lower than those of the microlenses fabricated without using the inverted air bath heating reflux method. The results show that this method can achieve high-precision surface control in microlens array fabrication.

## Figures and Tables

**Figure 1 micromachines-15-00289-f001:**
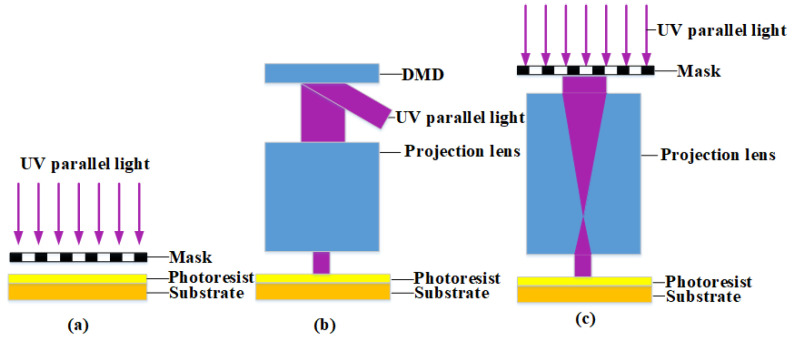
Schematic diagram of light paths for (**a**) contact and proximity lithography, (**b**) digital maskless lithography, and (**c**) projection lithography.

**Figure 2 micromachines-15-00289-f002:**
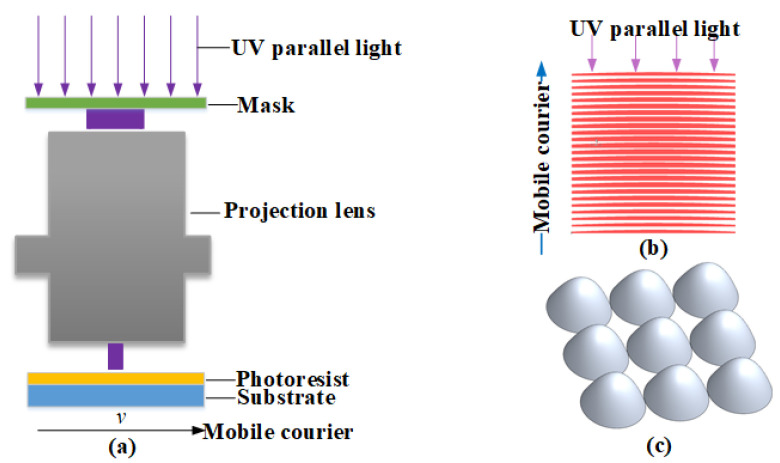
Mask-moving exposure principle. (**a**) Mask-moving exposure principle based on projection lithography, (**b**) design mask version graphic structure, and (**c**) the fabricated 3D nanostructure.

**Figure 3 micromachines-15-00289-f003:**
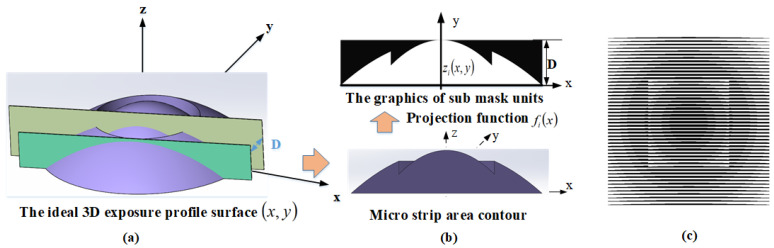
(**a**) Principle of the mask-moving filtering method for equally dividing the objective function, (**b**) converting the 3D structure in the equally divided area into a 2D curve and generating mask patterns of the divided areas based on it, and (**c**) combining the mask patterns of all the divided areas in turn to generate the shift-filtering mask of the objective function.

**Figure 4 micromachines-15-00289-f004:**
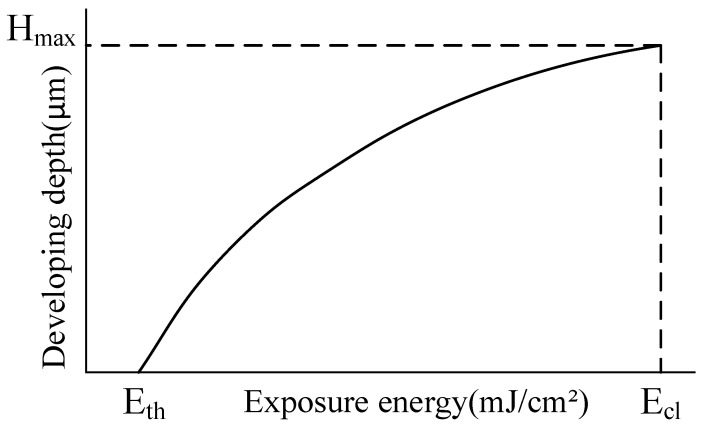
Relationship model between exposure dose and depth.

**Figure 5 micromachines-15-00289-f005:**
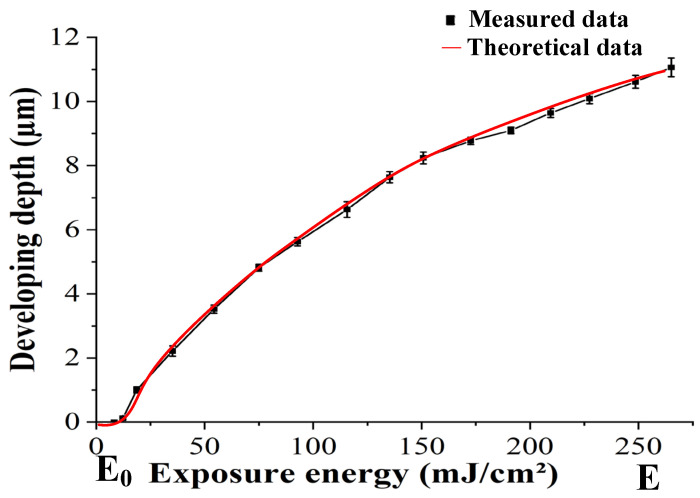
Relationship between exposure dose and exposure depth.

**Figure 6 micromachines-15-00289-f006:**
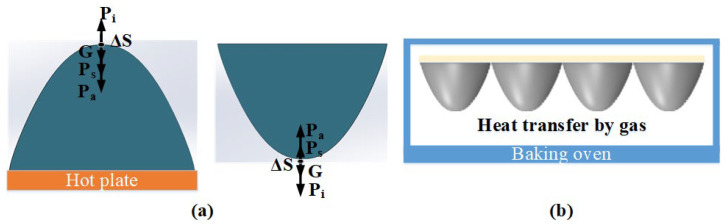
Comparison of the two reflux models. (**a**) Force analysis and comparison of two heat reflux modes. (**b**) Diagram of inverted gas bath reflux method.

**Figure 7 micromachines-15-00289-f007:**
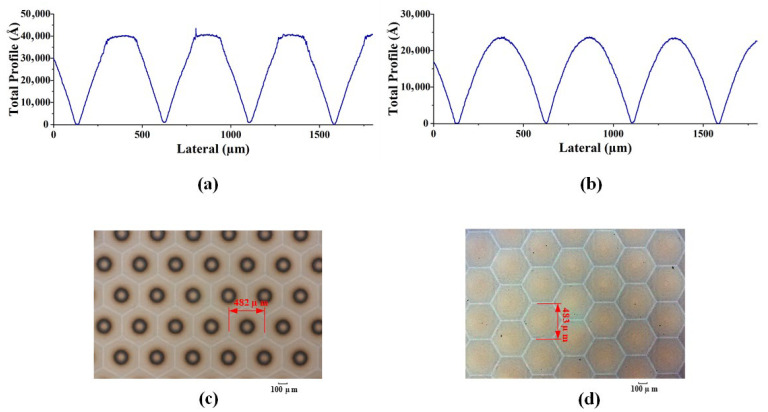
Machining results of hexagonal microlens. (**a**). Surface shape measurement results of conventional exposure. (**b**) Surface shape measurement results of front base pre-exposure. (**c**) Micrograph with conventional exposure. (**d**) Micrograph of front base pre-exposure.

**Figure 8 micromachines-15-00289-f008:**
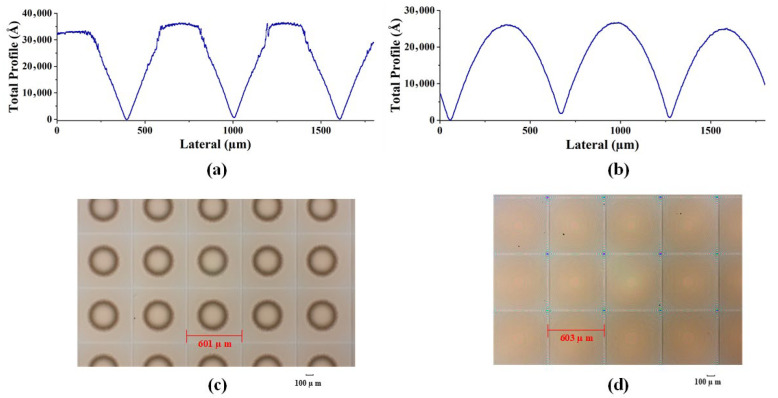
Machining measurement results of square-edged microlens. (**a**) Surface shape measurement results of conventional exposure. (**b**) Surface shape measurement results of front base pre-exposure. (**c**) Micrograph with conventional exposure. (**d**) Micrograph of front base pre-exposure.

**Figure 9 micromachines-15-00289-f009:**
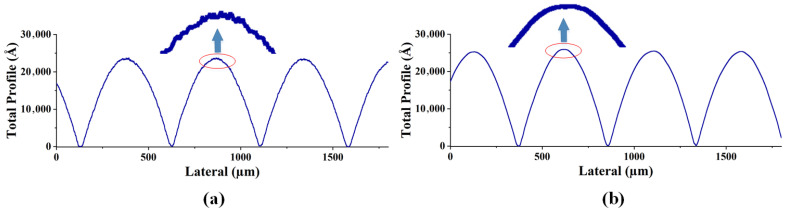
Surface shape measurements of the hexagonal microlens. (**a**) Surface shape measurements without the surface heating reflux method being used. (**b**) Surface shape measurements with the inverted air bath reflux method being used.

**Figure 10 micromachines-15-00289-f010:**
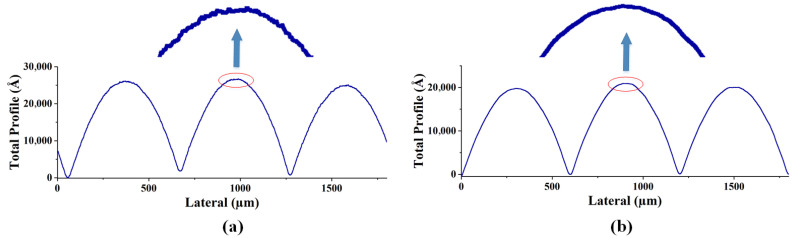
Surface shape measurements of the square microlens. (**a**) Surface shape measurements without the surface heating reflux method being used. (**b**) Surface shape measurements with the inverted air bath reflux method being used.

**Figure 11 micromachines-15-00289-f011:**
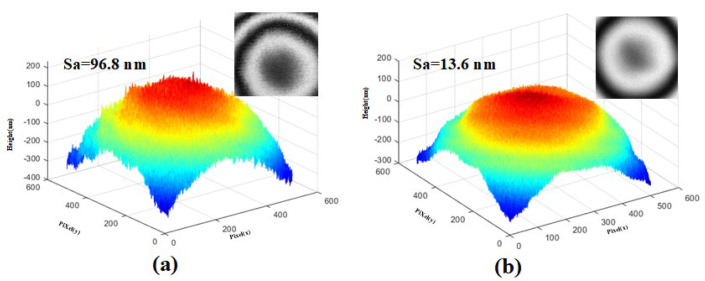
Morphology measurements of the hexagonal microlens: (**a**) morphology measurements without the surface heating reflux method being used and (**b**) morphology measurements with the inverted air bath reflux method being used.

**Figure 12 micromachines-15-00289-f012:**
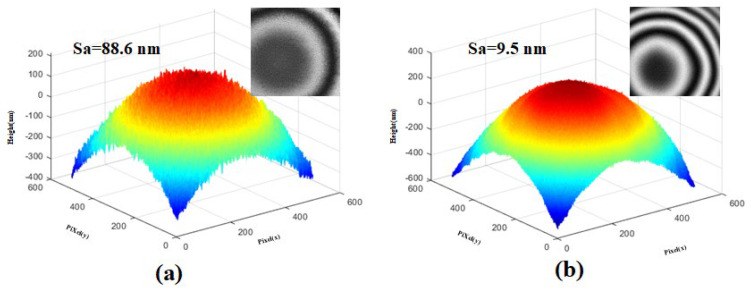
Morphology measurements of the square microlens: (**a**) morphology measurements without the surface heating reflux method being used and (**b**) morphology measurements with the inverted air bath reflux method being used.

**Table 1 micromachines-15-00289-t001:** Relationship between exposure dose and depth.

Exposure Dose (mJ/cm²)	Exposure Depth (µm)	Standard Deviation (µm)
8.3	0	0
12.3	0.12	0.03
18.7	1.0	0.10
35.2	2.22	0.16
54.3	3.52	0.13
74.9	4.81	0.11
92.8	5.62	0.13
115.6	6.63	0.24
135.3	7.64	0.17
150.8	8.24	0.18
172.6	8.77	0.11
191.2	9.1	0.10
209.5	9.64	0.14
227.4	10.09	0.16
248.6	10.61	0.20
265.1	11.2	0.27

## Data Availability

Data are contained within the article.

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
