# Peer review of "Mask-Moving-Lithography-Based High-Precision Surface Fabrication Method for Microlens Arrays"

_micromachines, 2024, doi:10.3390/mi15020289_

Round 1

Reviewer 1 Report

Comments and Suggestions for Authors

The authors propose an optimization technique for parpering high-precision microlens arrays. The pre-exposure technology can increase the development speed and not reduce the development threshold of the photoresist.The photoinitiator is the key for determining the threshold of the photoresist.Inthismanuscript, the inverted air bath reflux method isfunnyskillforeliminating surface protrusions via the microlens array surface to a molten state edge partofit.

(1)   The first word in picturesis always capitalized.

(2)   In page 2, itistosuggestthatthecontentofthe paragraphshouldbeillustrated in conjunction with Figure 1,andwhich is followed by the similar problem.

(3)   InFigute 2 and Figure 6, whatare(a) and (b),respectively?

(4)   In Figure 5, the coordinate axisshould have data and error bars.

(5)   Inpage 6, “In the microlens fabrication process, the photoresist does not undergo any decomposition in areas with an exposure dose less than the exposure threshold, leading to severe deformation in those areas.” Usually, no photocrosslinking reaction occurs at the location of the positive photoresist without light,so why the "severe deformation", and how to pove that.

(6)   Before submitting a revision be sure that your material is properly prepared and formatted, especiallythe references, and the quality of English needs improving.

Comments on the Quality of English Language

The authors propose an optimization technique for parpering high-precision microlens arrays. The pre-exposure technology can increase the development speed and not reduce the development threshold of the photoresist.The photoinitiator is the key for determining the threshold of the photoresist.Inthismanuscript, the inverted air bath reflux method isfunnyskillforeliminating surface protrusions via the microlens array surface to a molten state edge partofit.

(1)   The first word in picturesis always capitalized.

(2)   In page 2, itistosuggestthatthecontentofthe paragraphshouldbeillustrated in conjunction with Figure 1,andwhich is followed by the similar problem.

(3)   InFigute 2 and Figure 6, whatare(a) and (b),respectively?

(4)   In Figure 5, the coordinate axisshould have data and error bars.

(5)   Inpage 6, “In the microlens fabrication process, the photoresist does not undergo any decomposition in areas with an exposure dose less than the exposure threshold, leading to severe deformation in those areas.” Usually, no photocrosslinking reaction occurs at the location of the positive photoresist without light,so why the "severe deformation", and how to pove that.

(6)   Before submitting a revision be sure that your material is properly prepared and formatted, especiallythe references, and the quality of English needs improving.

Author Response

(1) The first word in pictures is always capitalized.

Re:I apologize for our carelessness. The description of the figure has been revised in the resubmitted manuscript.

(2)   In page 2, it is to suggest that the content of the paragraph should be illustrated in conjunction with Figure 1, and which is followed by the similar problem.

(3)  In Figure 2 and Figure 6, what are(a) and (b), respectively?

Re:Thanks for your advice. We have re-added the figure descriptions in the resubmitted manuscript. Figure 2(a) shows the principle of mask movement exposure based on projection lithography. Figure 2(b) shows the designed mask pattern structure. Figure 6(a) shows the force analysis and comparison of the two thermal reflow methods. Figure 6(b) shows the schematic diagram of the inverted gas bath reflux method.

(4)   In Figure 5, the coordinate axis should have data and error bars.

Re:The Figure 5 has been revised in the resubmitted manuscript. The data and error bars have been added in this figure 5. Thanks for your advice.

(5)   In page 6, “In the microlens fabrication process, the photoresist does not undergo any decomposition in areas with an exposure dose less than the exposure threshold, leading to severe deformation in those areas.” Usually, no photocross linking reaction occurs at the location of the positive photoresist without light, so why the "severe deformation", and how to pove that.

Re:After the ideal glue for VLSI is developed, the unexposed parts remain unchanged and the exposed parts are completely peeled off, requiring a large depth contrast. The existence of energy threshold is beneficial to the production of binary devices. Continuous components have different photoresist requirements than binary components, where subtle surface relief is obtained by a delicate response to changing energy. Since the energy weight of the high-frequency part of the signal is often very low, the photoresist has a photosensitive threshold, and these relief details will be suppressed, resulting in graphic distortion, which is equivalent to passing through a low-pass filter.

(6)   Before submitting a revision be sure that your material is properly prepared and formatted, especially the references, and the quality of English needs improving.

Re:This manuscript has been revised. And the formatted of references has also been revised.

Reviewer 2 Report

Comments and Suggestions for Authors

Please clearly explain the difference compared with the previous paper (Ref 27). The fabrication technique seems the same. Is the difference only on heating of the microstructure to improve the surface roughness? This is not something new. 

Ref 27. Gong, J.; Zhou, J.; Sun, H.; Hu, S.; Wang, J.; Liu, J. Mask-shifting-based projection lithography for microlens array fabrication. Photonics. 2023, 10, 1135. DOI:10.3390/photonics10101135.

Comments on the Quality of English Language

NA

Author Response

Ref 27. Gong, J.; Zhou, J.; Sun, H.; Hu, S.; Wang, J.; Liu, J. Mask-shifting-based projection lithography for microlens array fabrication. Photonics. 2023, 10, 1135. DOI:10.3390/photonics10101135.

Re: Thank you very much for the reviewer's comments. When the feature size of the microstructure is sub-micron and below, the traditional mobile mask processing method based on proximity contact cannot meet the processing accuracy requirements, resulting in serious microstructure contour distortion. Therefore, we proposed a projection-based mask moving processing method in Reference 27. During the exposure process, the feature pattern is scanned and irradiated onto the substrate surface. The exposure image on the substrate surface is the same as the mask feature pattern but with a reduced size. Spatial image, thereby obtaining continuous exposure distribution of characteristic patterns on the surface of the substrate, and achieving continuous exposure of characteristic patterns. Using a 0.2x photolithography objective lens for exposure imaging, under the same circumstances, the microstructure outline size of the mask surface is 5 times that of ordinary close-contact photolithography equipment. In addition, when the mask and the substrate move relative to each other, the mask and the sample are not in contact at all, and the product yield and the life of the mask will be greatly improved. The experimental results verify the feasibility of the projection-based mask movement processing method.

This study proposes a high-precision surface shape processing control method for microlens arrays based on projection mask moving lithography, which achieves the preparation of high-precision surface shapes of microlens arrays with small lens surface shape errors and low surface roughness. By utilizing the threshold characteristics of the photoresist development process and combining it with pre-exposure technology to increase its development speed and avoid the impact of the exposure threshold on the surface shape of the microlens, high-precision control of the surface profile of the microlens is achieved. The inverted gas bath hot melt method is used to heat the surface of the developed photoresist microlens to a molten state, thereby reducing irregular burrs on the surface of the microlens and reducing its surface roughness.

Reviewer 3 Report

Comments and Suggestions for Authors

The manuscript presented the fabrication of microlens arrays by two different methods and compared the results by characterization of their profiles. It is logically organized into sections and offers a meaningful comparison of methods for Micromachines. There are, however , a number of points that require further clarification to avoid confusion to the reader, as well as a more thorough introduction of state of the art in the literature. Detailed comments are included in the attached document and should be carefully addressed before further consideration of the manuscript. 

Author Response

1、It should be clarified in the title and abstract that the method utilized photolithographic techniques. This is currently not clearly described.

Re: Thanks for your advice. We have modified this issue in the resubmitted manuscript.

2、Reference 7 is the same as reference 2. Please rectify. Moreover, the relation of the report to fabrication of microlens arrays is not clear.

The literature review on fabrication methods is missing specific examples of relevant high resolution large area nanoseeding and nanoimprint lithography, see

https://doi.org/10.1088/1361-6528/aae795

https://doi.org/10.1063/5.0040839

The introduction of specific examples from the literature is suggested to explain the state of the art of the methods and suggest future developments.

Re: Your suggestions are very instructive for this manuscript. These references have been cited in the resubmitted manuscript.

3、It is not clear which of the methods corresponds to each sub-figure. It is suggested to indicate each sub-figure of figure 1.

Re: I apologize for our carelessness. The sublot descriptions in Figure 1 have been added in this resubmitted manuscript.

4、The figure is presented before its reference in text. Introduction of the figure prior to the literature is suggested to fully inform the reader about the methods. Moreover, the method followed in this work is not clearly indicated.

Re: This manuscript has been revised and the photolithography method used in this article has been clarified.

5、It is not clear how the pattern in 2b is moving to form the lenses. Adding the step by step schematic of figure 3 can address this issue.

Moreover, it is not clear what is the structure of the bulky projection lens.

Re: The mask moving filtering technology based on projection lithography mentioned in this article first divides the three-dimensional target structure into many fine strip areas (Figure 3a), and then encodes the microstructure in each strip area (Figure 3b ). When the target structure equal intervals are small enough, the relief structure in the tiny area is approximately a cylinder. During the exposure process, the mask is moved at a uniform speed along the target function equal division direction (Figure 2b), and the exposure distribution generated by the mask encoding unit is superimposed on each other. The resist surface exposure distribution is formed (Figure 2c).

6、The preparation of the photoresist was not presented. A minimum of appropriate concentrations and solvents is required as reported in efficient photoresist methods, see https://doi.org/10.1016/j.mee.2009.10.043

Re: Thanks for your advice. This manuscript has been revised.

7、The theoretical data were presented in figure 4. It is suggested to remove figure 4 to avoid repetition. Moreover, the numbers are missing from the figure. Please rectify.

Re: Figure 4 is a qualitative analysis model diagram of the relationship between exposure dose and exposure depth in principle; Figure 5 is a qualitative comparison between measured data and theoretical data. Specific data values have been added to the figure.

8、The caption did not describe what is presented in each subfigure. The distinction between hot plate and gas transfer is required.

Re: Figure 6(a) shows the stress analysis and comparison of the two thermal reflow methods, and Figure 6(b) shows the schematic diagram of the inverted gas bath reflux method.

9、The figure was presented before its reference in text. Conversion of A to um is suggested for a more clear presentation. The comments also apply to figures 8, 9, 10, 11 and 12

Re: The ordinate of the original measurement data of the step meter used in this article is angstroms. At the same time, compared with the lens aperture size, the sagittal height is relatively small. Using angstroms as the ordinate can more intuitively present the surface shape accuracy of the lens.

10、The maker and model of the step and 3D profiler is missing from the manuscript. Please add and explain.

Re: Thanks for your advice. An introduction to specific models of test instruments has been added in this resubmitted manuscript.

Round 2

Reviewer 3 Report

Comments and Suggestions for Authors

The authors applied detailed revisions to the comments of the review. The concerns were addressed. Please ensure all replies to comments are reflected in the main text of the manuscript.

Author Response

The authors applied detailed revisions to the comments of the review. The concerns were addressed. Please ensure all replies to comments are reflected in the main text of the manuscript.

Re: Thanks for your advice. We have carefully checked the revised content of the revised manuscript in accordance with the revision comments. And We ensure all replies to comments are reflected in the main text of the manuscript.